



# Variability of the properties of the distribution of the relative humidity with respect to ice: Implications for contrail formation

Sidiki Sanogo[1], Olivier Boucher[1], Nicolas Bellouin[1,2], Audran Borella[1], Kevin Wolf[1], and Susanne Rohs[3]

[1]Institut Pierre–Simon Laplace, Sorbonne Université / CNRS, Paris, France
[2]Department of Meteorology, University of Reading, Reading, United Kingdom
[3]Forschungszentrum Jülich GmbH, Institute of Energy and Climate Research 8 – Troposphere, Jülich, Germany

**Correspondence:** Sidiki Sanogo (sidiki.sanogo@ipsl.fr) and Olivier Boucher (olivier.boucher@ipsl.fr)

**Abstract.**

Relative humidity with respect to ice (RHi) is a key variable in the formation of cirrus clouds and contrails. We document its probability density function (PDF) using long-term Measurement of Ozone and water vapour on Airbus aircraft In-service programme (MOZAIC) and the In-service Aircraft for a Global Observing System (IAGOS) observations over the period 1995-

2022 in the upper troposphere (UT) and lower stratosphere (LS) between 325 hPa and 175 hPa. The characteristics of the RHi PDF differ in the UT and in LS of the high-latitudes (HL) and mid-latitudes (ML) regions of the Northern Hemisphere. In the LS, the probability ($P$) of observing a certain RHi decreases exponentially with increasing RHi. The rate of this decrease in $P$ with increasing RHi is greater in supersaturated than in subsaturated conditions. In the UT, $P$ first increases exponentially under subsaturated conditions then decreases exponentially in supersaturated conditions. Because of these different behaviours, the

PDF for the combined UT and LS is bimodal. In contrast to the HL and the ML regions, $P$ in the tropical troposphere decreases exponentially with increasing RHi. The different forms of PDF, in the tropics and in the higher latitude regions (ML and HL), lead to a global PDF of RHi in subsaturated tropospheric conditions that is almost uniform. This PDF shows a weak mode in the vicinity of 100 %, which can be associated essentially with the presence of cirrus clouds. These different characteristics of the RHi PDF exhibit some differences depending on the pressure level. These findings invite caution when using MOZAIC

and IAGOS measurements to calibrate large-scale simulations of RHi. The variability of RHi properties associated with that of the temperature has implications for the formation of contrails. We examined the impact of switching from the current fuel of aircraft, kerosene, to bio-ethanol, or to liquid-hydrogen on the frequency of contrails using the Schmidt-Appleman criterion. We show that bio-ethanol and more so hydrogen would produce more contrails. The impact of a potential change from kerosene to one of these two alternative fuels decreases with the decreasing pressure level but increases when moving from the high-

latitudes of the Northern Hemisphere to the tropics. We recommend that the comparison between models and observations be performed regionally and for the UT and LS separately. Finally, we emphasize that investigations on the impact on the contrail occurrence of switching from fossil kerosene to more sustainable fuels must be carried out in various climatic conditions.



# 1  Introduction

Water vapour is a trace gas in the Earth's atmosphere (Meerkötter and Vázquez-Navarro, 2012) but is responsible for two-thirds
of the natural greenhouse effect (Gierens et al., 2012). It is also responsible for one-sixth of the energy transport from Earth's
surface into the atmosphere via evapotranspiration of water at the ground and condensation and freezing in the atmosphere
(Gierens et al., 2012). A variable used to analyze the condensation and freezing processes at play in in situ cirrus clouds and
contrails formation is the Relative Humidity with respect to ice (RHi). Cases where RHi is 100 % are labelled "saturated". In
the upper troposphere (UT) and lower stratosphere (LS) region (UTLS) and also sometimes near the surface in Antarctica, RHi
can be ice supersaturated (RHi > 100 %) (Gierens et al., 1999; Gierens and Brinkop, 2012; Genthon et al., 2017; Petzold et al.,
2020).

Ice supersaturation occurrence is a prerequisite for the formation of in situ natural cirrus and contrail-cirrus (Schumann,
1996; Heymsfield et al., 2017). In situ cirrus clouds may form in the UT and in the LS via homogeneous nucleation at temper-
ature colder than –38 °C and ice supersaturations above 140 % are then necessary (Kanji et al., 2017; Heymsfield et al., 2017).
Cirrus clouds may also form via heterogeneous freezing at temperatures lower than 0 °C with ice supersaturation above 100 %
in the presence of ice nucleating particles (Kanji et al., 2017; Heymsfield et al., 2017). The formation mechanisms of contrails
are quite different to those of natural cirrus. They are linked to the atmospheric conditions that influence the complex processes
occurring in the wake of aircraft (Schumann, 1996; Kärcher, 2018). Schmidt (1941) and Appleman (1953) provided a simple
thermodynamic criterion, known as the Schmidt–Appleman criterion (SAc), which bypasses the complex dynamical processes
and only based on ambient temperature and relative humidity. It has been amended by Schumann (1996), taking into account
fuel combustion properties and the aircraft engine propulsion efficiency (see Sect. 2.4 for a refresher on the criterion).

Contrail-cirrus have a radiative forcing that is of relevance for the Earth's climate (Kärcher, 2018; Schumann et al., 2021;
Lee et al., 2021). They interact with both solar and terrestrial radiation and the net effect is a warming of the troposphere
(Lee et al., 2021). However, there is a large spread in the net radiative forcing for individual contrail cirrus and the average
magnitude of this effect is still uncertain. For instance, for the year 2018, the Effective Radiative Forcing (ERF) of contrail-
cirrus in high-humidity regions was estimated to be 57.4 mW m$^{-2}$ with a 5-95 % likelihood range of 17 to 98 mW m$^{-2}$. This
non-$CO_2$ forcing of aviation is stronger than that from $CO_2$ (34.3 mW m$^{-2}$). Overall, the uncertainty associated with the ERF
of contrails and contrail-cirrus represents a significant part of the uncertainty associated with the total ERF of aviation on the
Earth's climate (Lee et al., 2021). Natural cirrus clouds play an important role in the Earth's radiative budget (Fusina et al.,
2007; Boucher et al., 2013).

Accurate representation of the Ice SuperSaturation Regions (ISSR) in numerical weather prediction models is important to
improve the prediction of contrail-prone conditions, which is important for mitigation strategies aiming to reduce the climate
impact of the aviation sector through contrail avoidance (Sperber and Gierens, 2023). This first requires a good characterization
of the spatial and temporal distribution of RHi. For this purpose, the characterization of ISSR (corresponding to the upper tail of
the RHi distribution) has been carried out in some past studies (e.g., Petzold et al., 2020; Gierens et al., 1999). ISSR distributions
have been shown to depend on the location and season. In terms of spatial variability, the highest frequency is observed in deep





convection regions of the tropics above 200 hPa (Spichtinger et al., 2003b; Lamquin et al., 2012; Spichtinger et al., 2003a). ISSR are more frequent at 400-300 hPa and 300-200 hPa in the mid and high-latitude regions, respectively (Lamquin et al., 2012). In terms of seasonality, their highest frequency of occurrence is observed in winter, followed by spring, while they are

less frequent in summer and autumn (Petzold et al., 2020; Spichtinger et al., 2003b; Lamquin et al., 2012; Wolf et al., 2023). It should be noted that the magnitude of the ISSR frequency depend on the observational products being considered (e.g., airborne or spaceborne observations) due to differences in observation methods and their horizontal and vertical resolutions. Analysis using the European research programme MOZAIC showed that in the troposphere an aircraft above 300 hPa has a 20-30 % and 35-40 % probability of encountering an ISSR in summer and in winter respectively when flying over North America, the

North Atlantic and Europe (Petzold et al., 2020). So, the average spatial distribution of ISSR is fairly well documented in the literature. However, the occurrence of contrails depend also on the ambient temperature, the fuel combustion properties and the engine propulsion efficiency (Schumann, 1996).

The Probability Density Function (PDF) of RHi has also been analysed in several studies (e.g., Diao et al., 2014; Lamquin et al., 2012; Smit et al., 2014; Petzold et al., 2020) based on in situ measurements. They all found a bimodal PDF with a first

mode between 0 and 10 % and a second mode between 95 and 112 %. However, there is still a gap in our knowledge on the properties of tropospheric and stratospheric PDFs of RHi. Gierens et al. (1999) showed that the large scale PDF of RHi in the UT and LS are different at 250 and 200 hPa. The PDF of RHi in the LS follows an exponential decay in both ice supersaturation and subsaturation, and therefore does not show any break in slope in the vicinity of 100 %, unlike in the UT. A local scale study of Spichtinger et al. (2003a) over Lindenberg (Germany) reported that like in the UT, RHi PDF in the LS exhibits a break in

slope in the vicinity of 100 %. It should also be noted that the question of how these properties vary between the tropics and the mid and high-latitudes regions, and also with pressure level, has so far not been fully answered.

The objectives of this study are twofold: 1) to document the properties of the PDF of RHi in the UT and in the LS, in both clear and cloudy conditions as a function of the latitude and pressure, 2) to document using the Schmidt-Appleman criterion the occurrence frequency of conditions favorable to the formation of non-persistent and persistent contrails and the impact of

a fuel change on these frequencies.

This article is structured as follows. Section 2 details the dataset used in the present study and describes our methodology. Section 3 documents the properties of the PDF of RHi and the occurrence frequency of the conditions favorable to contrail formation. Finally, Section 4 summarizes our main findings.

## 2   Data and methods

### 2.1   Dataset from IAGOS and MOZAIC passenger aircraft

We analyzed RHi from airborne measurements spanning the period 1995 to 2022. Measurements of ambient temperature ($T$) and pressure ($p$), relative humidy with respect to liquid water (RHl), ozone volume mixing ratio ($m_{0_3}$), and ice crystal number concentration ($N_i$) are also analysed either to understand the properties of the PDF of RHi or to document the frequency of the conditions favorable to the formation of contrails. They are all obtained from the Measurement of Ozone and water vapour



on Airbus aircraft In-service (MOZAIC) programme (Marenco et al., 1998) over the period 1995-2014 and from the In-service Aircraft for a Global Observing System (IAGOS) programme (Petzold et al., 2015) over the period 2011-2022. Data for $N_i$ are available only from a subset of the IAGOS flights. The MOZAIC and IAGOS data are measured with a temporal sampling of 4 seconds. In this study, for each flight we used the latest version (amongst versions 3.1.1 to 3.1.4) data available at the time of our study.

Figure 1 shows the spatio-temporal distribution of RHi measurements between 325 hPa and 175 hPa defined here as the UTLS region. The best sampled areas are in the Mid-Latitudes (ML) of the northern Hemisphere. They are the Eastern United States of America (40° N – 60° N, 120° W – 65° W, USA hereafter), Europe (40° N – 60° N, 5° W – 30° E, EU hereafter) and especially the North Atlantic corridor (40° N – 60° N, 65° W – 5° W, NA thereafter). The sampling over Russia, Asia, the High-Latitudes (HL) of the northern Hemisphere and in the tropics is less dense. Our study is focused on the HL, ML, and the tropics. The areas of the ML spanning from USA to EU is analysed more in detail. The geographical coordinates of the different study areas are provided in Table 1. For documenting the PDF of RHi, we considered data above 325 hPa then split the UTLS into layers 325-275 hPa, 275-225 hPa and 225-175 hPa using a resolution of 50 hPa. Most measurements are performed in 275-225 hPa and 225-175 hPa (Fig. 2). Those performed in 325-275 hPa are less prominent (Fig. 2) but they represent 186812, 4963539, 2980377 measurements in the HL, the ML, and in the tropics, respectively.

**Table 1.** Names of study areas and their geographical coordinates.

| Names | Latitude and longitude ranges |
|---|---|
| High-Latitudes (HL) | 60° N – 80° N, 120° W – 150° E |
| Mid-Latitudes (ML) | 40° N – 60° N, 120° W – 150° E |
| Eastern United States of America (USA) | 40° N – 60° N, 105° W – 65° W |
| North Atlantic (NA) | 40° N – 60° N, 65° W – 5° W |
| Europe (EU) | 40° N – 60° N, 5° W – 30° E |
| Tropics | 25° S – 25° N, 100° W – 150° E |

## 2.2 Data selection

MOZAIC and IAGOS data are assigned different quality levels, ranging from "good" to "not validated". In this study, only "good" measurements of RHi, RHl, $T$, $p$, $m_{0_3}$, and $N_i$ are considered. A grounding problem with the IAGOS data acquisition system affected RHi between 2011 and 2017. During this period, RHi values are selected using the RHl quality flag. We followed Gierens et al. (1999) by separating UT air mass measurements and LS air mass measurements on the based of their $m_{0_3}$. The mean value of the ozone mixing ratio $m_{0_3}$ at the thermal tropopause being 130 ppb with a standard deviation of 92 ppb (Duhnke et al., 1998), we consider that measurements are from a UT air mass if $m_{0_3} < 130$ ppb and a LS air mass otherwise. This approach is used for a fairly comparaison of our results to those of Gierens et al. (1999). It is worth noting that a sensitivity analysis using the threshold of 2 potential vorticity unit data of the fifth generation of the European





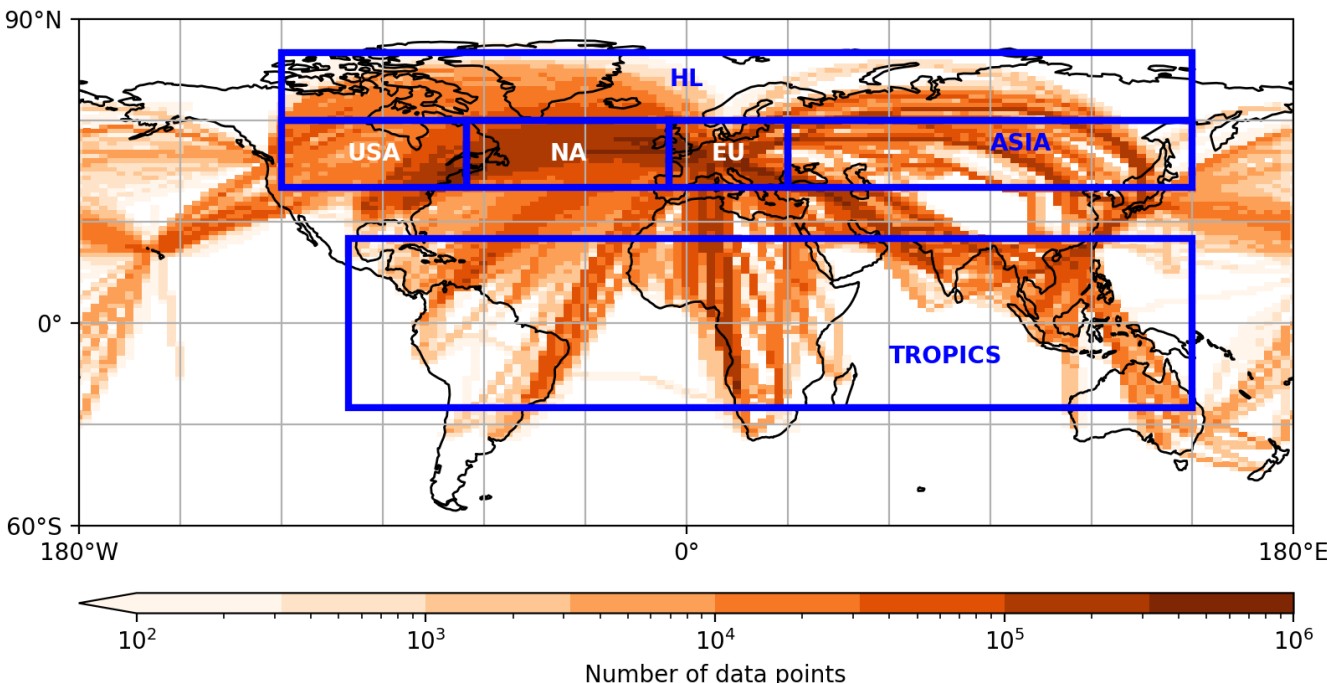

**Figure 1.** Global coverage of RHi observations by MOZAIC and IAGOS aircraft over the 1995-2022 period, between 325 hPa and 175 hPa shown as the total number of measurements per $2.5° \times 2.5°$ gridbox (log scale). The areas delimited by blue boxes represent the study zones. Their coordinates are provided in the Table 1.

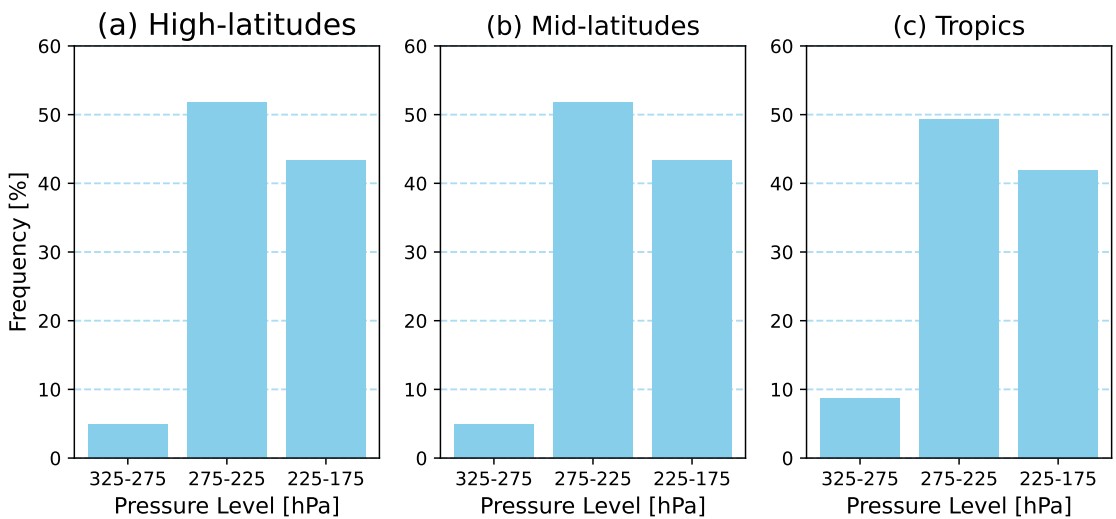

**Figure 2.** Frequency of RHi measurements per range of pressure level in the a) high-latitudes, b) mid-latitudes of the Northern Hemisphere and c) in the tropics over the period 1995-2022.



Centre for Medium-Range Weather Forecasts atmospheric reanalysis (ERA5) to discriminate tropospheric and stratospheric
measurements showed consistent results with the $m_{0_3}$ based approach (not shown).

## 2.3 Differentiation of clear sky and cloudy conditions

The $N_i$ variable is used to differentiate data measured in cirrus clouds from those in clear sky conditions. Different thresholds
are used in the literature. For instance Beswick et al. (2015) and Lloyd et al. (2020) used a threshold of 0.05 particles cm$^{-3}$
while Petzold et al. (2017) used a lower threshold of 0.015 particles cm$^{-3}$. Here we used an even smaller threshold of 0.001 par-
ticles cm$^{-3}$, i.e, we considered as in cloud measurements those with $N_i \geq 0.001$ particles cm$^{-3}$. We used this threshold to
be more restrictive on the discrimination of the clear conditions for the investigations of the origin of the wet mode in the
PDF of RHi. We then used the thresholds of 0.015 particles cm$^{-3}$ and 0.05 particles cm$^{-3}$ for sensitivity test since a detection
uncertainties that can exceed 50 % is associated with the threshold of 0.001 cm$^{-3}$ (Petzold et al., 2017). To exclude potential
measurements in the presence of supercooled liquid water, as in Petzold et al. (2017), we considered only $N_i$ data for which the
temperature is colder than –40 °C, which corresponds to the threshold for the spontaneous freezing of water droplets (Petzold
et al., 2017). Due to low sampling of HL by IAGOS aircraft equipped with $N_i$ measurement sensors, the analysis involving
screening clear and cloudy conditions is restricted only to the ML and to the tropics.

## 2.4 Contrail detection: Schmidt-Appleman criterion

The combustion of kerosene (or alternative fuels) releases hot and humid air behind the aircraft, which is progressively diluted
in the cold and dry ambient atmosphere. This mixture follows a line in a $p-T$ diagram (Schumann, 1996). If RHi of the mixture
is subsaturated, while $T$ is colder than a critical ambient temperature $T_{crit}$ and RHl is higer than a critical value RHl$_{crit}$, the
non-persistent contrails form (Schumann, 1996). Contrails are persistent if RHi is ice supersaturated (See Fig. 3 in Schumann,
1996). We used this criterion known as SAc to document the frequency of non-persistent and persistent contrails formation as a
function of the pressure level for the following fuels: kerosene, bio-ethanol (ethanol, hereafter) and liquid-hydrogen (hydrogen,
hereafter).

The approximation of $T_{crit}$ frequently used in the literature (e.g., Rap et al., 2010) is given by the following equation:

$$T_{crit} = 226.69 + 9.43 \cdot \ln(G - 0.053) + 0.7272 \cdot \ln^2(G - 0.053) \tag{1}$$

where $G$ (in Pa K$^{-1}$) is the slope of the mixture line in $p-T$ diagram and is defined as follows:

$$G = \frac{EI_{H_2O} \cdot c_p \cdot p}{\epsilon \cdot Q(1 - \eta)} \tag{2}$$

$G$ combines the atmospheric properties (the ambient pressure (in Pa) of the flight altitude, the isobaric heat capacity of air
$c_p$ = 1004 J kg$^{-1}$ K$^{-1}$, and the ratio of the molecular masses of water vapour and dry air $\epsilon \approx 0.622$), the fuel properties (the
specific combustion heat of the fuel $Q$ (in J kg$^{-1}$) and the emission index of water vapour for the fuels $EI_{H_2O}$ (in kg kg$^{-1}$))



and the aircraft engine propulsion efficiency of the aircraft $\eta$. In this study, we consider $\eta = 0.3$ which corresponds to a typical present-day fleet value (Schumann, 1996, 2012).

The values of $Q$ and $EI_{H_2O}$ for the different fuels are listed in Schumann (1996) and in Wolf et al. (2023).

The $RHl_{crit}$ threshold is determined by:

$$RHl_{crit} = \frac{G \cdot (T - T_{crit}) + e_{sat}^{liq}(T_{crit})}{e_{sat}^{liq}(T)} \tag{3}$$

where $e_{sat}^{liq}$ is the saturation water vapour pressure. For more details, the reader should refers for example to Schumann (1996) and Rap et al. (2010).

## 3 Results

### 3.1 Number of ice crystals in cirrus clouds

Values of RHi that are ice supersaturated and subsaturated are observed in both clear and cloudy conditions (Kahn et al., 2009; Krämer et al., 2009, 2016; Petzold et al., 2017). Therefore, for a better characterization of the PDF of RHi, we first document how clear and cloudy conditions are sampled in IAGOS measurements. For this purpose, we used three different detection
thresholds (0.001, 0.015, 0.05 particles cm$^{-3}$) for characterizing clouds, based on their ice crystal number concentration $N_i$ (see Sect. 2.3). We found, consistently with Petzold et al. (2017), that IAGOS aircraft encounter cirrus clouds with larger $N_i$ in the tropics than in the ML (Fig. 3). 63-56 % and 47-30 % of the observed non-zero $N_i$ in cirrus clouds are respectively greater than 0.015 particles cm$^{-3}$ and 0.05 particles cm$^{-3}$, in the pressure range 275-175 hPa in the tropics (Fig. 3). In the ML, the percentage of $N_i \geq 0.015$ particles cm$^{-3}$ and $N_i \geq 0.05$ particles cm$^{-3}$ in 325-275 hPa and in 275-225 hPa are 58 % and 40 %,
respectively (Fig. 3). The highest fraction of $N_i < 0.015$ cm$^{-3}$ in ML is observed at the altitudes of 225-175 hPa. It represents 72 % of the observed $N_i$ (Fig. 3). The higher $N_i$ in cirrus clouds in the tropics compared to the ML between 275-175 hPa is consistent with the findings of studies using synergetic lidar-radar satellite data (e.g., Sourdeval et al., 2018) and can be explained by the fact that, in the tropical region, strong updrafts in convective regions producing high ice supersaturation cause high nucleation rates that lead to high number concentrations of ice crystals (Krämer et al., 2016).

### 3.2 How long do IAGOS aircraft fly in cirrus clouds?

The fractions of aicraft flying time in cirrus clouds are presented in Table 2. They are computed for each pressure range as the number of measurements of $N_i$ collocated with measurements of temperature lower than –40 °C that are at least equal to the cirrus detection threshold divided by the total number of measurements. These fractions can also be interpreted as cirrus occurrence frequencies as seen by IAGOS.
In 275-225 hPa and 225-175 hPa pressure ranges, the occurrence frequency of cirrus clouds decreases in the ML with pressure level while it increases in the tropics. Using 0.001 particles cm$^{-3}$ as a threshold, we determined fractions of 4.4% and





**Table 2.** Number of total measurements (collocated with measurements of temperature lower than –40 °C) of the ice crystals number concentration ($N_i$) and the percentage of total measurements, for which $N_i \geq 0.001$ particles cm$^{-3}$, $N_i \geq 0.015$ particles cm$^{-3}$ and $N_i \geq 0.05$ particles cm$^{-3}$ in the mid-latitudes (ML) of the Northern Hemisphere and in the tropics over the period 2011-2022.

| Regions | Mid-latitudes | | | Tropics | |
|---|---|---|---|---|---|
| Pressure ranges (hPa) | 325-275 | 275-225 | 225-175 | 275-225 | 225-175 |
| Total number of measurements with T < –40 °C | 155620 | 2905616 | 4402016 | 1539032 | 4502899 |
| Fraction of measurements ($N_i \geq 0.001$ particles cm$^{-3}$) | 6.0 % | 3.7 % | 1.8 % | 4.4 % | 8.2 % |
| Fraction of measurements ($N_i \geq 0.015$ particles cm$^{-3}$) | 3.5 % | 1.4 % | 0.5 % | 2.7 % | 4.6 % |
| Fraction of measurements ($N_i \geq 0.05$ particles cm$^{-3}$) | 2.0 % | 0.7 % | 0.2 % | 2.0 % | 3.3 % |

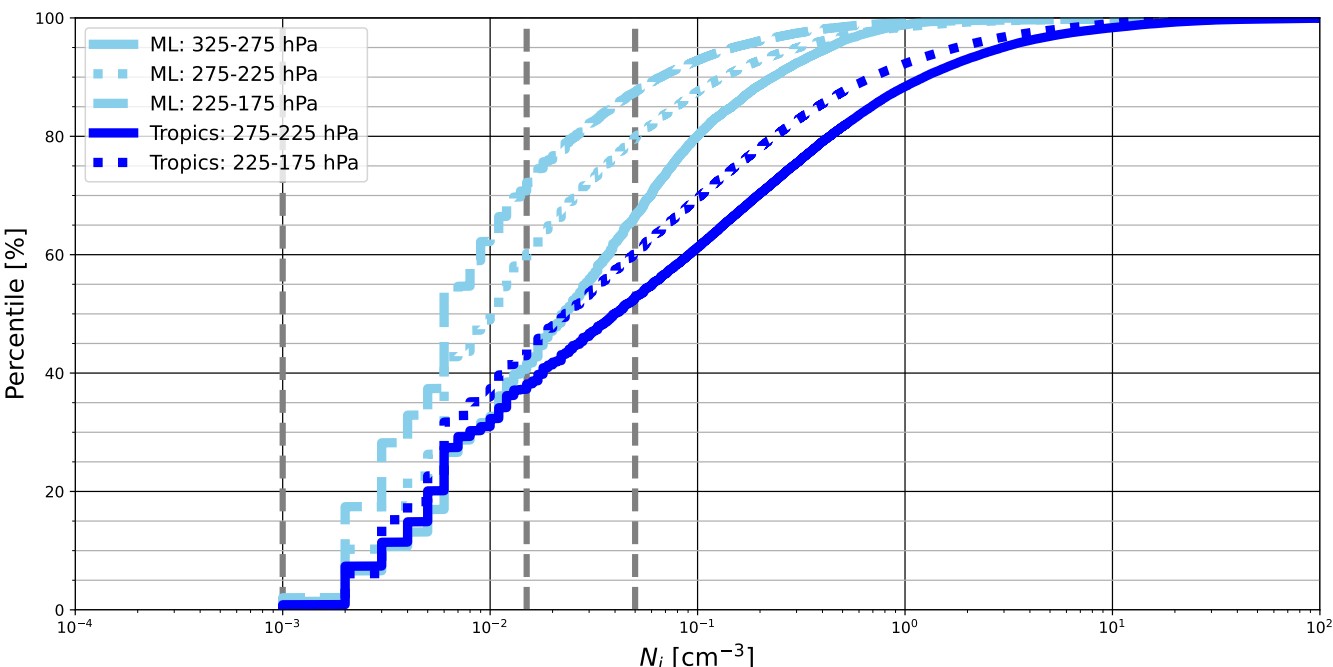

**Figure 3.** Ice crystal number concentrations in the mid-latitudes (ML) of the Northern Hemisphere and in the tropics over the period 2011-2022. The vertical dashed gray lines correspond to the concentration thresholds of 0.001 cm$^{-3}$, 0.015 cm$^{-3}$ and 0.05 cm$^{-3}$.

8.2% in the tropics and 3.7 % to 1.8 % in the ML (Table 2). On average, the total occurrence frequency of cirrus is highest in the tropics (Table 2). This is consistent with the findings of Petzold et al. (2017) who showed that IAGOS data capture the global pattern of cirrus clouds. When the detection threshold is increased to 0.015 particles cm$^{-3}$ and 0.05 particles cm$^{-3}$, the cirrus clouds occurrence frequency decreases, in particular in the 225-175 hPa layer (Table 2), but remains relevant. This suggests that the properties of RHi PDF from MOZAIC and IAGOS data are a combination of clear-sky and cloudy-sky conditions.




### 3.3 Characteristics of RHi probability density function

#### 3.3.1 Upper troposphere

Gierens et al. (1999) showed that over larger spatial and longer timescales, the distributions of subsaturated and supersaturated
RHi in the UT follow a uniform law and an exponential law, respectively. RHi is however modulated in space and time by
absolute humidity and temperature fluctuations (Diao et al., 2014). Consequently, statistical distribution of RHi will be subject
to spatial variability. This is illustrated in Fig. 4 that shows the PDF of RHi in the ML, the HL and in the tropics, for the
pressure ranges 325-275, 275-225 and 225-175 hPa. In the ML, the characteristics of the RHi PDF depend on the pressure
level. It is unimodal in 325-275 hPa with a mode at 108 % (Fig. 4d) while it is bimodal in the 275-225 and 225-175 hPa layers.
The two modes in 275-225 hPa are centered at 5 % and 108 % (Fig. 4e) while those in 225-175 hPa are located at 8 % and
100 % (Fig. 4f). The existence or the magnitude of the dry mode in the 275-225 and 225-175 hPa layers in the ML is not
certain. It may correspond to LS measurements, erroneously attributed to the UT because of uncertainties in the method for
discriminating UT and LS measurements of RHi. We therefore focused the following analysis on the RHi values greater than
25 %. It should be noted that the presence of a wet mode is reported in the early study of Spichtinger et al. (2003a) based on
radiosoundings over Lindenberg (Germany). In agreement with Reutter et al. (2020) who analyzed RHi in the North Atlantic,
the mean and the standard deviation of RHi distributions in the UT vary little with the pressure level (Table 3).

An important point, common to these UT RHi PDF and consistently with Gierens et al. (1999), is that the probability ($P$) of
observing a certain RHi beyond a supersaturated value $S$ (here, $S \in [100\ \%, 110\ \%]$) decreases exponentially with increasing
RHi. This exponential distribution can be written as follows (Spichtinger et al., 2003a):

$$P(RHi) = \lambda \cdot \exp(-\lambda \cdot RHi) \tag{4}$$

where $\lambda$ is the rate parameter.

For a quantitative intercomparison of the part of the RHi PDF corresponding to RHi $\geq S$, we take $S = 100\ \%$, and as Gierens
et al. (1999), we fit each PDF using the following straight line:

$$\ln P(RHi) \approx a + b \cdot \text{RHi} \tag{5}$$

in which $a$ is the intercept and $b$ is the slope. For the three different pressure intervals, $b$ (multiplied by 100 for easier reading) is
close to 4 and $a$ is between 2 and 3 in the ML (Table 4). The PDF of subsaturated values of RHi have different characteristics.
The probability $P$ of observing RHi $\in [25\ \%, 100\ \%[$ increases ($b \geq 0$), instead of decreasing exponentially with increasing RHi
in the 275-225 and 225-175 hPa layers (Table 4). In 325-275 hPa, the PDF is nearly uniform ($b \approx 0$) for RHi $\in [25\ \%, 100\ \%[$
(Table 4). Comparing USA, North Atlantic, and Europe revealed that, at the first order, the distribution of RHi over the three
regions are consistent with each other for the 275-225 and 225-175 hPa layers (Fig. 5). However, substantial differences appear
for 325-275 hPa. The magnitude of the dry mode at these pressure levels is higher in the USA compared to the other two
areas. In addition, the probability of observing RHi between 25 % and 100 % is almost uniform over Europe, while it increases



and decreases exponentially over the Nord Atlantic and USA, respectively (Fig. 5). This implies differences between the three
sub-regions in the mechanisms that modulate RHi variability in the pressure range 325-275 hPa. Weak sampling may also be a
contributing factor, since fewer measurements were made at these pressure levels (Fig. 2).

The UT RHi PDF in the HL exhibit similar features to those in the ML (Fig. 4a-c). For RHi between 25 % and 100 %, the
exponential increasing rate $b$ is slightly higher for the HL than the ML (Fig. 4a-f, Table 4).

In the tropical region, the evolution of the PDF of the RHi with pressure level is different (Fig. 4g-i). The PDFs are bimodal
and in contrast to the ML and the HL, the magnitude of the dry mode decreases with pressure level while that of the wet
mode increases. This is due to the warm tropical temperatures, which favours low RHi values for 325-275 hPa despite the
higher availability of water vapour compared to the other two pressure ranges considered here. Unlike ML and the HL, the
mode between 0 % and 25 % is probably real in the tropics since the measurements analyzed are less affected by stratospheric
measurements. For RHi $\geq$ 25 %, the PDF is characterised by an exponential decay on both sides of the mode at 100 %. It
should however be noted that the characteristics of the PDF beyond 100 % are similar to those of the RHi PDF in the ML
and HL. The important difference is that the absolute value of the rate parameter $b$ decreases with decreasing pressure in the
tropics, whereas it increases in ML and HL (Table 4). It is worth noting that the global PDF of RHi includes these opposing
tendencies in the probability of occurrence of RHi values lower than 100 % between the tropics and the Northern Hemisphere
into an almost uniform PDF ($b \approx 0$, Fig. 4j-l). These results are consistent with the findings of Gierens et al. (1999).

**Table 3.** Long-term (1995-2022) mean and standard deviation of the tropospheric and stratospheric (in brackets) in the High and Mid-
Latitudes of the North Hemisphere (denoted ML and HL respectively) and in the tropics.

| RHi PDF parameters | Mean (%) | | | Standard deviation (%) | | |
|---|---|---|---|---|---|---|
| Regions | High-Latitudes | Mid-Latitudes | Tropics | High-Latitudes | Mid-Latitudes | Tropics |
| 325 - 275 hPa | 88.9 (33.0) | 71.5 (31.3) | 27.0 | 38.6 (23.9) | 33.7 (23.1) | 35.2 |
| 275 - 225 hPa | 48.6 (24.4) | 78.4 (28.4) | 34.2 | 48.6 (23.7) | 32.2 (25.4) | 18.8 |
| 225 - 175 hPa | 56.9 (30.7) | 69.8 (24.5) | 38.8 | 56.9 (22.5) | 36.4 (24.7) | 33.8 |

### 3.3.2 Lower stratosphere

In this section, we document the RHi PDF in the LS. This analysis is carried out only in the ML and the HL since stratospheric
measurements are very few in the 325-175 hPa layer in the tropics. Mean RHi decrease with decreasing pressure level in the
ML and HL while the standard deviation varies very little (Table 3). Reutter et al. (2020) found similar results over the North
Atlantic Ocean.

Since the observed temperature in the UT and the LS are relatively close, while little water vapour crosses the tropopause
(Reutter et al., 2020; Petzold et al., 2020), the mean value of RHi in the LS is lower than in the UT. Another important
difference with the troposphere, common to the two regions and to the three pressure ranges considered here, is that the
probability of observing a specific RHi greater than 25 % decreases exponentially with increasing RHi (Fig. 4a-f, Table 5).





**Figure 4.** Probability density function of the relative humidity with respect to ice (in %) computed over the period 1995-2022 for the pressure ranges 325-275 hPa, 275-225 hPa, and 225-175 hPa, a-c) in the High-Latitudes (HL) of the Northern Hemisphere, d-f) in the Mid-Latitudes (ML) of the Northern Hemisphere (ML), g-i) for the tropical region, j-l) the PDF of the three domains combined. For each pressure range, the tropospheric RHi PDF (blue), the stratospheric RHi PDF (red) and their combined PDF (black) are shown. The green dashed and dotted lines correspond to the fit for RHi ∈ [25 %, 100 %[ and for RHi ∈ [100 %, 170 %[, respectively, in the stratosphere. The violet dashed and dotted lines are those for the troposphere.



**Figure 5.** Same as Fig. 4 but for the Eastern North America (USA), the North Atlantic (NA), and Western Europe (EU).





**Table 4.** Values (with uncertainty range of $\pm$ 1 standard deviation) of the parameters $a$ and $b$ of the fit line (see Eq. 5) of the RHi supersatured values (top) and the subsaturated values (bottom), computed over the period 1995-2022. The values of $b$ are multiplied by 100 for the sake of legibility.

| Fit parameters | $a$ | | | $100 \cdot b$ | | |
|---|---|---|---|---|---|---|
| Regions | High-Latitudes | Mid-Latitudes | Tropics | High-Latitudes | Mid-Latitudes | Tropics |
| 325 - 275 hPa | $2.3 \pm 0.24$ | $2.1 \pm 0.12$ | $4.3 \pm 0.48$ | $-3.7 \pm 0.18$ | $-3.7 \pm 0.09$ | $-6.1 \pm 0.04$ |
|  | $-2.9 \pm 0.04$ | $-2.0 \pm 0.01$ | $-1.9 \pm 0.02$ | $1.21 \pm 0.07$ | $0.00 \pm 0.01$ | $-0.45 \pm 0.00$ |
| 275 - 225 hPa | $2.7 \pm 0.16$ | $2.3 \pm 0.10$ | $3.7 \pm 0.18$ | $-4.1 \pm 0.011$ | $-3.8 \pm 0.07$ | $-5.4 \pm 0.04$ |
|  | $-2.7 \pm 0.01$ | $-2.4 \pm 0.01$ | $-1.8 \pm 0.02$ | $0.09 \pm 0.01$ | $0.06 \pm 0.02$ | $-0.48 \pm 0.00$ |
| 225 - 175 hPa | $3.8 \pm 0.29$ | $2.7 \pm 0.13$ | $2.0 \pm 0.09$ | $-5.1 \pm 0.21$ | $-4.3 \pm 0.01$ | $-3.9 \pm 0.04$ |
|  | $-2.7 \pm 0.02$ | $-2.4 \pm 0.00$ | $-1.8 \pm 0.01$ | $0.08 \pm 0.03$ | $0.04 \pm 0.00$ | $-0.41 \pm 0.00$ |

This exponentially decaying PDF, with a break in the slope around 100 %, is a feature reported by Spichtinger et al. (2003a) over Lindenberg (Germany). Our results show that this property is common to the RHi PDF in the ML and the HL of the
Northern Hemisphere. Gierens et al. (1999) did not find the break in slope around 100 % in the MOZAIC data for the period 1995-1997. This might be due to an undersampling of the LS properties of RHi PDF over this period. It is the case here for the stratospheric PDF of RHi in the 325-275 hPa layer ($b \approx 0$, Fig. 4a,e, Fig. 5a,d,g). It is important to note that, similarly to the UT, the probability of observing a supersaturated RHi increases with decreasing pressure. It should also be noted that, unlike the UT, the dry mode between 0 % and 25 % in the LS is expected due to its low water vapour content (see Petzold et al.
(2020)).

**Table 5.** Same as Table 4 but for the lower stratosphere.

| Fit parameters | $a$ | | $100 \cdot b$ | |
|---|---|---|---|---|
| Regions | High-Latitudes | Mid–Latitudes | High-Latitudes | Mid-Latitudes |
| 325 - 275 hPa | $1.2 \pm 0.34$ | $-0.2 \pm 0.09$ | $-4.0 \pm 0.29$ | $-2.6 \pm 0.06$ |
|  | $-1.3 \pm 0.03$ | $-1.2 \pm 0.01$ | $-1.5 \pm 0.05$ | $-1.8 \pm 0.01$ |
| 275 - 225 hPa | $1.4 \pm 0.16$ | $0.8 \pm 0.04$ | $-4.1 \pm 0.12$ | $-3.6 \pm 0.04$ |
|  | $-1.6 \pm 0.00$ | $-1.5 \pm 0.04$ | $-1.3 \pm 0.01$ | $-1.3 \pm 0.00$ |
| 225 - 175 hPa | $1.7 \pm 0.18$ | $1.8 \pm 0.13$ | $-4.4 \pm 0.14$ | $-4.4 \pm 009$ |
|  | $-1.8 \pm 0.01$ | $-1.8 \pm 0.01$ | $-1.2 \pm 0.02$ | $-1.2 \pm 0.01$ |

### 3.3.3 Lower stratosphere and upper troposphere

As shown above, there are differences between the shapes of RHi PDF in the UT and in LS in the HL and in the ML. In this section, we analyzed the properties that emerge when UT and LS are not separated. Such an approach is used in several





studies (e.g., Lamquin et al., 2012; Smit et al., 2014; Petzold et al., 2020; Diao et al., 2014). We found, in agreement with these
studies that RHi PDF in UTLS is bimodal (Fig. 4a-f). The dry mode of the PDF of RHi between 0 % and ∼50 % (depending
on pressure range and region) is essentially a characteristic of the LS while that for RHi greater than ∼50 % is dominated
by the features of RHi PDF in the UT (Fig. 4a-f). The magnitude of the dry mode varies with pressure level. It increases
from 325 to 175 hPa in the LS (Fig. 4a-f) since at these pressure levels in the LS, water vapour content decreases whereas
temperature increases (Reutter et al., 2020; Petzold et al., 2020). It should be noted that the RHi PDF of the ML includes more
UT measurements than the RHi PDF of the HL since the pressure level of the tropopause increases (the altitude decreases)
with latitude. Consquently, the shape of the upper tail of the RHi PDF is more dominated by the tropospheric PDF in the ML
than in the HL (Fig. 5). In the ML, some differences can be noted at the sub-regional scale on the shape of the RHi PDF. The
exact ranges of the lower/upper tail of the PDF of RHi dominated by the LS/UT features RHi depend on the location (Fig. 5).

### 3.3.4   Clear vs cloudy conditions

To further document the RHi PDF, we compared its properties in clear and cirrus cloud conditions between 325-175 hPa in the
ML and in the tropics using the sub-sample of data measured onboard aircraft equipped with ice crystal number concentration
measuring sensor. In these two regions, the aforementioned wet mode, in the vicinity of 100 % in the RHi PDF that combines
clear and cloudy conditions comes essentially from cirrus clouds (Fig. 6). Conditions with $N_i < 0.001$ particles cm$^{-3}$ as-
sociated with the slow and complex processes of cirrus clouds formation and dissipation may also contribute. Consequently,
this mode is not completly inexistent in clear sky conditions depending on the region. In the subsample of the IAGOS data
analysed here, it is more prominent in the tropics than in the ML (Fig. 6a-c). Petzold et al. (2017) conducted similar analyses
in different regions including, areas in the tropics and ML regions using IAGOS data of the period from July 2014 to October
2015 and found that RHi wet mode in the tropical Atlantic is more prominent (see their Fig. 8b). This characteristic seems to
be smoothed out in satellite and radar-lidar synergistic data, since Kahn et al. (2009) and Lamquin et al. (2012) reported no
peaks in the vicinity of 100% under clear conditions.

The positions of the in-cloud RHi mode in the ML and in the tropics are different (Fig. 6a-c). It is subsaturated in the
tropics while it is supersaturated in the ML (Fig. 6a-b). Cirrus clouds exhibit ice-subsaturated and ice supersaturated conditions
depending on their state of life (Petzold et al., 2020; Li et al., 2023). Some studies (e.g., Petzold et al., 2017; Krämer et al.,
2009) have then found that the mode of the PDF of RHi in cirrus clouds is ice supersaturated while others (e.g., Li et al.,
2023; Ovarlez et al., 2002; Kahn et al., 2009) found an ice sub-saturated mode. Figure 6c shows that the subsample of IAGOS
flight data analyzed here sampled tropical cirrus clouds, characterized by essentially by undersaturated RHi. These cases may
correspond to cirrus clouds in the dissipation phase. However, since the level of supersaturation of this humid mode of RHi
depends on the region as illustrated in this study and reported in the literature, further studies are needed to better characterize
it and elucidate the mechanisms involved.





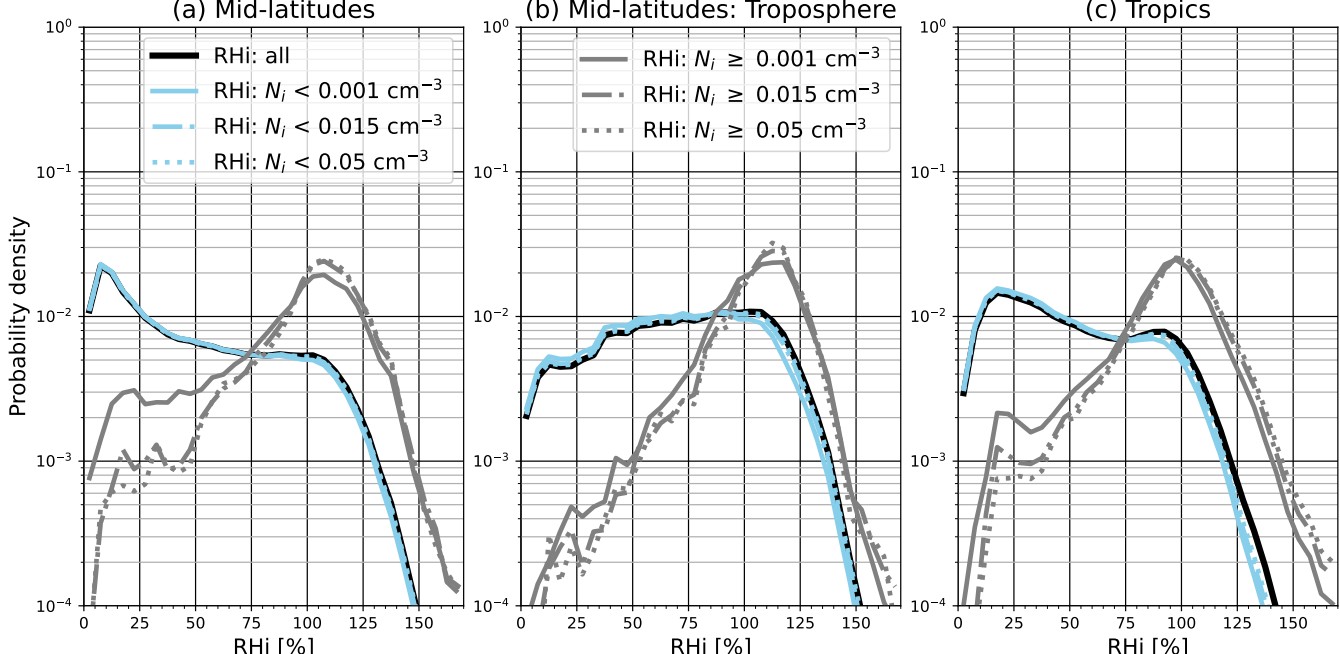

**Figure 6.** Probability density function of RHi in the clear conditions (sky blue) and cloudy sky (gray), and in the total conditions (black), a) in the upper troposphere and lower stratosphere in the mid-latitudes of the Northern Hemisphere, b) only in the upper troposphere in the mid-latitudes of the Northern Hemisphere and c) in the tropical upper troposphere and lower stratosphere, computed over the period 2011-2022. Only RHi measurements collocated with measurements of temperature lower than –40° are plotted.

## 3.4 Implications of RHi variability for contrail formation

### 3.4.1 Frequency of ice supersaturation regions

Subsequently, we investigate the frequency of occurrence of ISSR, which are necessary for persistent contrail formation (Schmidt, 1941; Appleman, 1953). We computed this frequency as the ratio of aircraft flight time in ISSR to total flight time. In the HL and in the ML, ISSR frequency decreases with the pressure level (Fig. 7a-c). In the HL, for 325-275, 275-225, 225-175 hPa, the occurrence frequencies are respectively 19.2 %, 11.0 %, 5.5 %, while they are 20.8 %, 17.6 % and 9.0 % in ML. In the tropics, ISSR occur more frequently (13.6 % of the time) in the pressure range of 225-175 hPa. In the 325-275 and 275-225 hPa layers they occur 7.9 % and 10.4 % of the time, respectively. It should be noted that, whether in the tropics or in the mid or high-latitudes regions, the frequency of ISSR is characterized by regional and seasonal variability which is well documented in the literature (see Sect. 1).







**Figure 7.** Frequency (%) of ISSR (in cyan) and of persistent contrail formation conditions for kerosene (sky blye), liquid ethanol (red) and hydrogen (blue) fuels over the a) high and b) mid-latitudes of the Northern Hemisphere and over c) the tropics.





**Figure 8.** Frequency (%) of non-persistent contrail formation conditions for with kerosene (sky blue), liquid ethanol (red) and hydrogen (blue) fuels over the a) high and b) mid latitudes of the Northern Hemisphere and over c) the tropics.



### 3.4.2 Frequency of contrail formation and fuel choice

Investigations are being carried out by the aviation industry on the possibility of using bio-ethanol (ethanol, hereafter) and liquid-hydrogen (hydrogen, hereafter) as an alternative fuel to kerosene. Ethanol and hydrogen are two of the so-called Sustainable Aviation Fuel (SAF). SAF use is expected to reduce aviation-induced $CO_2$ emissions if they are generated from carbon-neutral sources (Ng et al., 2021). They could however produce contrails which have a net warming effect on the climate (see Sect. 1). Here, we used the SAc to test the impact of using ethanol or hydrogen on the formation of contrails compared to kerosene. The SAc also accounts for the aircraft-engine propulsion efficiency $\eta$ (see Sec. 2.4) which is expected to continue to increase in the future (Sahoo et al., 2020). But, since our study is focused only on the impact of fuel choice, we used a typical present-day aircraft-engine propulsion efficiency of 0.3. The SAc is applied to MOZAIC and IAGOS measurements, varying the specific combustion heat of the fuel $Q$ and the emission index of water vapour for the $EI_{H_2O}$ of kerosene, ethanol, and hydrogen. The values of $Q$ and $EI_{H_2O}$ are set to 43.2 J kg$^{-1}$ and 1.25 kg kg$^{-1}$ for kerosene, to 27.2 J kg$^{-1}$ and 1.17 kg kg$^{-1}$ for ethanol, to 120 J kg$^{-1}$ and 8.94 kg kg$^{-1}$ for hydrogen (Schumann, 1996; Wolf et al., 2023).

The fraction of flying time during which aircraft can produce persistent contrails as function of the fuel used is shown in the Fig. 7. These fractions or frequencies depend on the region and the altitude of the flight. For instance, MOZAIC and IAGOS aircraft using kerosene and flying in the HL at the altitude, corresponding to the pressure range 325-275 hPa, have produced persistent contrails during 15.6 % of the flight time (Fig. 7a). This frequency decreases to 10.8 % and to 5.5 % for the 275-225 and 225-175 hPa layers, respectively. For 325-275 hPa, MOZAIC and IAGOS aircraft would have formed contrails more often if they had used hydrogen or ethanol (19.2 % and 17.8 %, respectively) instead of kerosene, while the fuel used has almost no effect for the 225-175 hPa layer (Fig. 7a). The comparison with the frequency of ISSR showed that for 325-275 hPa, with the kerosene aircraft fly in ice supersaturated air masses for 3.6 % of the time without producing persistent contrails (Table 6). This frequency is equal to 1.4 % for ethanol (Table 6) and 0 % for hyrogen indicating that with hydrogen fuel, persistent contrails would form as soon as the air masses are ice supersaturated. Similar conclusions are found for the 275-175 hPa layer (Table 6).

Like the HL, the impact on the formation of persistent contrails of switching from kerosene to ethanol or to hydrogen decreases with decreasing pressure level in ML. In this region, with kerosene as fuel, aircraft produce persistent contrails more frequently (for 15.94 % of the flight time) between 275-225 hPa (Fig. 7b) which correspond to the pressure ranges where they fly most (Fig. 2). This occurrence frequency is about 11.3 % between 325-275 hPa and 8.9 % between 225-175 hPa (Fig. 7b). The frequencies of persistent contrails induced by hydrogen closely follow the ISSR frequencies (Fig. 7b). Overall, the fuel switching impact is highest for 325-275 hPa and lowest for 225-175 hPa (Fig. 7b). It is only with hydrogen that persistent contrails form at almost all the pressure ranges considered here as soon as ISSR conditions occur (Table 6).

Persistent contrail occurrence as function of fuel in the tropics increases with decreasing pressure level. More importantly, the impact of switching from kerosene to ethanol or to hydrogen is most important where aircraft fly the most (275-225 hPa) (Fig. 7c, Fig. 2c). At these altitudes, persistent contrails occur 0.5 % of the time with kerosene but would occur 4.0 % and 9.5 % of the time, respectively, with ethanol and hydrogen (Fig. 7c). This impact is also important between 225-175 hPa (Fig. 7c).



**Table 6.** Difference between the frequency of persistent contrails associated with the kerosene, the bio-ethanol and the liquid-hydrogen and the frequency of ice supersaturation, in the High and Mid-latitudes of the North Hemisphere (denoted HL and ML respectively) and in the tropics, computed over the period 1995-2022.

| Regions | High-latitudes | | | Mid-latitudes | | | Tropics | | |
|---|---|---|---|---|---|---|---|---|---|
| Fuels | Kerosene | Ethanol | Hydrogen | Kerosene | Ethanol | Hydrogen | Kerosene | Ethanol | Hydrogen |
| 325 - 275 hPa | 3.6% | 1.4% | 0.0% | 9.5% | 4.9% | 1.2% | 7.9% | 7.9% | 7.1% |
| 275 - 225 hPa | 0.2% | 0.0% | 0.0% | 1.6% | 0.3% | 0.0% | 9.8% | 6.4% | 0.9% |
| 225 - 175 hPa | 0.0% | 0.0% | 0.0% | 0.0% | 0.0% | 0.0% | 3.5% | 0.4% | 0.0% |

The temperature in the UTLS is on average, colder in the HL and in the ML than in the tropics (Alder et al., 2011). The Clausius-Clapeyron relationship indicates that the saturation vapour pressure with respect to ice and with respect to liquid water increases with the temperature. That partly explains the variability with pressure level and latitude of switching from kerosene to ethanol or hydrogen. Combustion releases hot and humid air behind the aicraft, which is progressively diluted to the cold and dry ambient atmosphere. Consequently, the fuel whose having the higher ratio $EI_{H_2O}$ and $Q$ (here hydrogen and ethanol), assuming that the other parameters of the combustion remain equal, tends to form more persistent contrails at the highest temperatures.

The frequency of non persistent contrail formation increases with latitude and the pressure level between 325 and 175 hPa (Fig. 8). At these pressure layers, the impact of switching from kerosene to ethanol or hydrogen and is prominent in the tropics (Fig. 8). Our results extend those of Wolf et al. (2023), who showed that the impact of switching from kerosene to ethanol or to hydrogen will be greater on the occurrence of non-persistent contrails than on persistent contrails, in the Paris region.

## 4   Summary and conclusions

In the present study, we documented the properties of RHi over the period 1995-2022 using the long-term MOZAIC and IAGOS observations onboard passenger aircraft. The frequency of the contrail formation conditions is also analysed. The analyses are carried out over the high-latitudes (HL) and mid-latitude (ML) regions of the Northern Hemisphere and in the tropics, in the upper troposphere (UT) and lower stratosphere (LS), between 325 and 175 hPa separated into three pressure ranges with a resolution of 50 hPa. RHi properties are also documented in clear sky and in cirrus cloud conditions. Measurements with ozone volume mixing ratio below 130 ppb were flagged as have been measured in the UT, while those corresponding to ozone volume mixing ratio above 130 ppb are flagged as LS measurements. We used the thresholds 0.001, 0.015 and 0.05 particles cm$^{-3}$ of the ice crystal number concentration $N_i$ for cloud detection. The contrail formation conditions have been detected using the Schmidt-Appleman criterion considering an aircraft-engine propulsion efficiency of 0.3. The main results are the following:

1. The cirrus cloud fraction sampled by IAGOS aircraft depends on the detection threshold and the location. The highest cirrus cloud frequency is observed in the tropics. This frequency can be interpreted as the IAGOS aicraft flying time



in cirrus clouds. With the detection threshold of 0.001 particles cm$^{-3}$, the frequencies are 4.4 % and 8.2 % at aircraft cruising altitude for the 275-225 and 225-175 hPa layers, respectively. In the ML, these respective frequencies are respectively 3.7 % and 1.8 %. They are of the order of 2.7 % and 4.6 % in the tropics, 14 % and 0.5 % in the ML for the cirrus having at least a concentration of ice crystals number of 0.015 particles cm$^{-3}$. These frequencies are slightly

lower if the detection threshold of 0.05 particles cm$^{-3}$ is used. Cirrus clouds are characterized by higher amount of ice supersaturation than clear sky, which shifts the upper tail of the overall RHi PDF towards more ice supersaturation with a peak around 100 %.

2. We characterized the PDF of RHi in the UT. In the HL, the ML and the tropics, the probability of observing a certain RHi decreases exponentially, with increasing RHi in ice supersaturation conditions. In sub-saturated conditions, the

probability of observing a certain RHi increases exponentially with RHi in the HL and the ML regions while it decreases exponentially in the tropics. Combining these different shapes of subsaturated RHi PDF in the tropics and in mid and high latitudes (ML and HL) lead to a global tropospheric RHi PDF that follows a almost uniform distribution.

3. The LS RHi PDF is different from that in the UT in the HL and ML. The probability of observing a given RHi decreases exponentially with increasing RHi in both sub- and supersaturated regions. However, the decrease slope of the distri-

bution is higher under saturated conditions than subsaturated conditions. The combination of these different shapes of subsaturated RHi PDF in the LS and in the UT leads a total bimodal PDF of RHi in the ML and in the HL.

4. The probability of forming non-persistent and persistent contrails by aircraft using bio-ethanol and liquid hydrogen instead of kerosene are analysed. Both are candidate fuels for reducing the climate impact of aviation. We found that these alternative fuels are more likely to produce contrails than kerosene. However, the magnitude of their impact on

persistent contrail formation depends on pressure level and latitude. In the HL of the Northern Hemisphere, switching from kerosene to liquid-hydrogen or bio-ethanol is more likely to have very little impact on the persistent contrails frequency for the 275-225 hPa and 225-175 hPa layers. In this region, aircraft form persistent contrails nearly every time they encounter air masses that are supersaturated with respect to ice at these altitudes. The same conclusions are found for 225-175 hPa in the ML. Overall, the impact of switching from kerosene to hydrogen or bio-ethanol on contrail

occurrence decreases with the pressure level between 325 and 175 hPa, and from the tropics to high-latitudes region. The impact of switching from kerosene to hydrogen will be more important in for non-persistent contrails.

This study updates and completes some aspects of the studies of Gierens et al. (1999) and Spichtinger et al. (2003a) on the characteristics of the RHi PDF. It emphasizes that observation and model comparisons need to be performed for UT and LS separately, and preferentially over different regions and pressure levels, distinguishing cloud-free and cloudy conditions.

Models that are calibrated to reproduce the global RHi PDF from MOZAIC and IAGOS data risk doing so for the wrong reasons. Similarly, observing systems designed to monitor UTLS humidity will need to be able to distinguish UT from LS. This may be a challenge for satellite-based systems, which may not have a sufficiently good vertical resolution. Finally, studies on the impact on the contrail occurrence of switching from fossil kerosene to more sustainable fuels must be conducted in various climatic conditions.



*Data availability.*  IAGOS data are available from the IAGOS data portal (https://doi.org/10.25326/20). We use IAGOS data accessed on 3 november 2023.

*Author contributions.*  SS, OB and NB designed the study. SS carried out the analysis and the preparation of the manuscript. OB and AB helped with the analysis. OB, AB, NB, KW contributed to the preparation of the manuscript. SR provided the MOZAIC and IAGOS data.

*Competing interests.*  The authors declare that they have no conflict of interest.

*Acknowledgements.*  S. Sanogo, O. Boucher, N. Bellouin, A. Borella and K. Wolf acknowledge support from the French Ministère de la Transition écologique (grant no. DGAC 382 N2021-39), with support from France's Plan National de Relance et de Resilience (PNRR) and the European Union's NextGenerationEU. We acknowledge the strong support of the European Commission, Airbus and the airlines (Deutsche Lufthansa, Air France, Austrian, Air Namibia, Cathay Pacific, Iberia, China Airlines, Hawaiian Airlines, Eurowings Discover, and Air Canada) that have carried the MOZAIC or IAGOS equipment and performed the maintenance since 1994. IAGOS has been funded by
the European Union projects IAGOS–DS and IAGOS–ERI. Additionally, IAGOS has been funded by INSU-CNRS (France), Météo-France, Université Paul Sabatier (Toulouse, France) and Research Center Jülich (FZJ, Jülich, Germany). The IAGOS database is supported in France by AERIS (https://www.aeris-data.fr)



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
