# Peer review of "Variability in the properties of the distribution of the relative humidity with respect to ice: Implications for contrail formation"

_EGUsphere, 2023_

## Referee Comment (RC2)

**Comments on "Variability of the properties of the distribution of the relative humidity with respect to ice: Implications for contrail formation" by Sanogo et al.**

**General Comments:**

This study by Sanogo et al. updates and extends a number of studies that have analysed the large MOZAIC and IAGOS data sets. Specifically, the PDFs of RHi have been derived for the period 1995 – 2022 for the upper troposphere (UT) and the lower stratosphere (LS) separately as well as combined for different geographical regions, three pressure layers and cloudy and clear-sky conditions. The authors find that the PDFs differ substantially between the UT and the LS as well as between cloudy and clear-sky conditions. Moderate differences appear between different geographical regions and pressure layers. Furthermore, the authors analysed the occurrence frequencies of ISSRs and of persistent and non-persistent contrails for different combustion fuels. Here, they also find differences between the considered geographical regions and pressure layers. The frequency of contrail occurrence increases from kerosene over bio ethanol to liquid hydrogen.

Overall, the study contributes to the detailed documentation and analysis of the relative humidity field at common cruise altitudes and provides a valuable basis for the evaluation of the quality of atmospheric models. This fits into the scope of ACP and is of scientific interest, in particular since NWP models need to improve in order to make operational contrail mitigation feasible. Thus, the manuscript is suitable for publication in ACP.

However, the quality of the manuscript is substantially lowered by a lack of mathematical and physical precision in its formulation as well as deficiencies in the use of the English language. Therefore, the manuscript should be revised considerably, before it can be published. Please note that I focused my review on scientific issues and, beyond this, only commented on typos. However, I also noticed numerous language errors including additional or missing words, unsuitable words or the inappropriate use of singular/plural. I recommend that the authors go through their manuscript sentence by sentence to improve on this and may consider to consult a native speaker if necessary.

**Specific Comments:**

- The abstract appears a bit too detailed to me, especially between lines 6 and 14. Please consider summarizing the results more concisely.
- Abstract, line 7, lines 192f, 195, 202, 207, 222, 232, 348, 350, 353: "the probability (P) of observing a certain RHi" is mathematically unprecise as the following sentences are clearly describing the behaviour of the probability density. Specifically, "the probability of observing a certain RHi" is always zero as the boundaries for integrating the PDF are identical in this case. I suggest sticking to the terms "PDF" and "probability density" and not denoting the PDF with "P", as this is the common symbol for the probability.
- Abstract, line 19, lines 226, 308, 314, 325, 365, "decreasing pressure level": Unclear what you mean. Apparently, you sometimes mean increasing and sometimes decreasing pressure by that throughout the text. I suggest leaving out "level" and just using "increasing/decreasing pressure".
- Line 28: "saturated" → "saturated with respect to ice"

- Lines 29, 130ff, 152, 179, 238, 271: I am sure what you mean is that the "water vapour" is ice supersaturated. "RHi" may rather be above 100% or above saturation.
- Lines 34f: In the common definition of supersaturation, 140% ice supersaturation would refer to RHi = 240%. Suggestion: "ice supersaturation" → "RHi"
- Page 2, third paragraph: I missed a sentence on recent research suggesting a low efficacy for contrail cirrus potentially resulting in a low temperature response despite the high ERF (e.g. Bickel 2023).
- Line 51 f: An accurate representation of ISSRs is important for the prediction of contrail persistence, not "contrail-prone conditions" in general.
- Line 58: The way the sentence is written, it seems to state that ISSR are more frequent between 400 and 300 hPa in the midlatitudes and between 300 and 200 hPa in the high latitudes, which is not the case when looking at figure 9 in Lamquin et al. (2012). Please switch the order of either the pressure levels or the latitude regions.
- Line 101: As the three pressure ranges appear numerous times in the paper, you may consider labelling them with something like "lower", "intermediate" and "upper" layer to improve the readability.
- Line 104: Is the number of measurements in the HL missing a digit or is it really an order of magnitude lower than in the other latitude regions on this pressure level?
- Line 108: Why is this a good replacement? Are the quality flags of RHi and RHl correlated during periods, where the "grounding problem" did not exist?
- Lines 130f: Not RHi of the mixture but RHi in the ambient atmosphere has to be below 100% for non-persistent contrails. I suggest that you leave out "If RHi of the mixture is subsaturated," and simply state that contrails in general form under these conditions, as you mention the criterion for contrail persistence in the next sentence anyway. Also, please specify that you mean the properties of the ambient atmosphere by "T" and "RHl".
- Lines 156-160: Unclear description of fig. 3. I cannot reproduce the 30% for the 0.05 threshold in the tropics from the plots. Also, you mention two thresholds for the ML, but only give the percentages for one of them.
- Line 191: Apparently, this is only true for the ML.
- Lines 236f, "It is the case here for the stratospheric PDF of RHi in the 325-275 hPa layer (b ≈ 0, Fig. 4a,e,…": Unclear. What is the case? Under sampling? Also, b is clearly not zero for the stratosphere and figure 4e does not correspond to the mentioned layer.
- Line 237f: Hard to verify this from the plots. The opposite seems to be the case for example for EU.
- Figure 6: The PDF of in-cloud RHi increases for subsaturated conditions and decreases for supersaturated conditions when decreasing the detection threshold, indicating that the measurements that are added in this process were mainly sampled under subsaturated conditions, some of these under strongly subsaturated conditions, as even a "dry mode" appears in the PDFs. You briefly mention "detection uncertainty" in section 2.3, but never come back to this issue. Could these ice crystals in very dry air be crystals that have been turbulently mixed out of cirrus nearby or even false measurements? In both cases, these measurements would not really correspond to cloudy conditions. Please discuss your choice of threshold.
- Line 306: I don't see, how the results for the other pressure layers are similar, as the frequencies are (almost) zero there, regardless of the fuel.
- Line 311f: Repetition of line 307f
- Line 318: Please be more precise what you mean. This is certainly only true when referring to the same pressure level. The tropical tropopause is usually colder than the tropopause in the

ML and HL, but it is also located at a much lower pressure such that the tropical UTLS is not properly represented in your data set.

- Page 19, first paragraph: Please rewrite the paragraph. Your argumentation is hard to follow. Try structuring it:
ISSR + low ambient temperature → mixture of exhaust and ambient air exceeds water saturation due to curvature of vapour pressure curve, regardless of fuel. → persistent contrail
ISSR + higher ambient temperature → mixture of exhaust and ambient air may stay subsaturated with respect to liquid water if ratio of $EI_{H_2O}$ to $Q$ is low. → no contrail with kerosene, persistent contrail with ethanol or hydrogen
- Line 330: "the properties of RHi" → "properties of the atmospheric RHi PDF"
- Line 334: "RHi" → "RHi PDF"
- Line 340f, "This frequency can be interpreted as the IAGOS aicraft flying time in cirrus clouds." This information belongs into the paragraph above, as this is not a result. Also. "estimated by" might be better than "interpreted as".
- Line 343: A dot is missing in "14 %". Anyway, I do not think this amount of details is necessary in a summary. "of the order of" implies that you give a rough estimate, but you repeat the exact numbers from the results section. I think, the interesting results are that cirrus is more frequent in the tropics than in the ML, that it occurs higher up in the tropics and that the frequency decreases if a higher detection threshold is chosen.
- Line 362: "In this region, aircraft form persistent contrails…" Suggestion: "The reason is that, in this region, aircraft running on kerosene already form persistent contrails…"
- Line 364: "contrail" → "persistent contrail", also, the lower level in the tropics seems to be an exception from this rule.

**Technical Corrections:**

- The formulation of the title seems a bit unwieldy to me. Suggestion: "Variability in the distribution properties of the relative humidity with respect to ice: Implications for contrail formation"
- Abstract, line 2, lines 89f: Several alternatives can be found on the internet for the meaning of the acronym MOZAIC. The IAGOS website itself gives the two possibilities "Measurement of OZone and water vapour on AIrbus in-service airCraft" and "Measurements of OZone and water vapour by in-service AIrbus airCraft". I suggest using one of these.
- Abstract, line 15: I suggest starting a new paragraph after "large-scale simulations of RHi.".
- Page 2, third paragraph: The last sentence seems a bit unconnected. I suggest inserting "also" after "Natural cirrus clouds".
- Line 87: Typo in "relative humidy"
- Line 112: Typo in "comparaison"
- Figure 2: The font size of "High-latitudes" appears to be larger than above the other two panels.
- Figure 3: It would be easier to compare the two latitude regions if equal pressure ranges were plotted with the same line style.
- Line 201: b is close to -4
- Line 208: Typo in "Nord"

- Table 3, caption: "RHi" is missing. Also, ML and HL have to be switched. Actually, this bracket is unnecessary anyway, as these abbreviations are not used in the table. The same holds for the caption of table 6.
- Figure 4: "ML" is redundant in line 3 of the caption. Also, the font size of "Probability Density" appears to be larger in the last row of images. Furthermore, I would not call the plots of the last row "global", as they are clearly not.
- Figure 5: Typo in the figure key: "LT"
- Table 5: A dot is missing in the bottom right cell.
- Line 252: The reference probably should be figure 4, not 5.
- Line 253: The second "RHi" is redundant.
- Line 257: Typo in "aformentioned"
- Line 260: Typo in "completly"
- Figure 6, caption: The "C" is missing behind the temperature value.
- Figure 7, caption: Typo in "blye"
- Figures 7 and 8: A grid or at least horizontal lines would improve the readability of the figures. Also, one might consider combining the two figures into one by plotting the non-persistent contrails on top of the persistent ones. This way, the frequency of non-contrail conditions could be read from the figure as well.
- Line 305: Typo in "hyrogen"
- Line 342: "respective" is redundant.
- Line 355: "saturated" → "supersaturated"

**Reference:**

Bickel, M. (2023). *Climate impact of contrail cirrus* (Doctoral dissertation, lmu).

---

## Author Comment (AC1)

**Response to Reviewer 1**

We would like to thank the anonymous Reviewer for their careful review. We found all the remarks and recommendations to be very relevant. Based on the review, we have extended the analyses to the seasonal scale, which has further enriched the paper. Below is our response to the comments on a point-by-point basis. The following convention for text fonts is used:

- Review comments in black color
- Our answers in blue color
- *Pieces of text taken from the revised manuscript in red color and in italic font.*

**Comments**

- Figure 5: As the regional differences here are sometimes quite significant (differences in dry mode), it would be interesting to know how much these results differ seasonally.

We thank the Reviewer for the remark. We took it into account by intercomparing the seasonal PDF of RHI over the USA, NA and EU for both the lower stratosphere (LS) and the upper troposphere (UT). We found that the difference between the three regions on the magnitude of the dry mode does not depend much on the season. We provided the analysis (Fig. S1) as Supplementary Material.  It should however be noted that restricting the dataset to collocated measurements that are flagged as valid for ozone, RHi and temperature and then separating the data between tropospheric and stratospheric measurements reduced the size of the seasonal samples. Consequently, all aspects of the seasonal PDF of RHi are not statistically robust. The comments about this analysis are on line 204 in the revised manuscript.

- Section 3.4:
  - Even if reference is made to the literature on the seasonality of ISSRs, I would also find a seasonal view or statement helpful for the results of Figures 7 and 8.

We thank the Reviewer for this very relevant remark. The frequency of occurrence of the persistent contrails closely follows that of ISSR. Consequently, it is the seasonality of the frequency of ISSR that we have documented. The values are presented in a new figure (Fig. 7 in the revised version of the manuscript). The comments added in the manuscript about this figure are on lines 285 to 302. A clarification is also made in the introduction from line 57 to line 60. As recommended, we have also provided a seasonal view of the impact of switching from kerosene to ethanol or hydrogen on the occurrence of contrails. The values are presented in Tables S1,S2,S3, in the Supplementary Material. The comment about these tables in the revised manuscript are on line 337. We have removed Table 6 in the manuscript because we found that it is redundant with the current Fig. 8.

  - In the explanation of the different contrail frequencies, it is mentioned that this is **partly** due to the ratio of EI_h2o to Q. What other reasons are there for behaviour?

The fuel-dependent parameters in the Schmidt-Appleman criterion equations are $EI_{H20}$ and Q. These two parameters are the main source of the differences in the frequency of contrails. Given that a switch of fuels would also likely induce changes in the aircraft design, the aircraft engine propulsion efficiency may also change and contribute to the differences. On the suggestion of Reviewer 2, we rephrase the explanation as follows: *At low ambient temperature, the contrail plume exceeds water saturation more readily due to the curvature of the vapour pressure curve, regardless of the fuel being considered. The contrail plume may stay subsaturated with respect to liquid water if the ratio of $EI_{H2O}$ to Q is low. In such conditions, persistent contrails are less frequent with kerosene than with hydrogen and ethanol.* This new text runs from the line 338 to the line 341 in the revised manuscript.

- ○ Could you briefly discuss how the results relate to climate projections. What is the expected impact of the different fuel types in a warming climate based on your results?

We now discuss how the results relate to climate projections between the lines 341 and 344 as follows: *With the warming of the upper troposphere expected as a result of climate change (Kumar et al. 2022), the impact of switching from kerosene to ethanol or hydrogen on contrail formation could further increase in the future, potentially affecting pressure layers (e.g., 225-175 hPa in the high and mid-latitudes) and seasons (winter and spring in the high and mid-latitudes) where it is low under present condition*s.

**Minor**

- P2L49 The last sentence in the paragraph sounds a bit tacked on

We have modified the paragraph by placing the statement about natural cirrus clouds at the beginning of the paragraph. Please find it on line 37 in the revised manuscript.

- Table 3: There is a word missing in the caption.

Corrected

- P14L271 remove the "by" between "essentially" and "undersaturated"

Done

- P18: Values for combustion heat Q should be in MJ /kg instead of J / k

Corrected

- P19L325 missing dash between "non" and "persistent"

Added

- P19L326 remove "and"

Done

---

## Author Comment (AC2)

**Response to Reviewer 2**

We would like to thank the Reviewer for their careful review. We found all the remarks and recommendations to be very relevant. As a result, we have improved the interpretation and presentation of our results, which, we hope, has significantly enhanced the quality of the paper. Below is our response to the specific comments on a point-by-point basis. The following convention for text fonts is used:

- Reviews comment in black color
- Our answer in blue color
- *Pieces of text taken from the revised manuscript in red color and in italic font.*

Imprecisions in mathematical and physical formulations reported in the general comments have been clarified. Finally, as recommended, we have revised the whole manuscript to improve the use of the English language.

**Specific Comments:**

1. The abstract appears a bit too detailed to me, especially between lines 6 and 14. Please consider summarizing the results more concisely.

Our original thinking was that the section on "impact of a fuel change on the occurrence of contrails" might be of interest to a large community (beyond the academic world). We had therefore presented the results in detail in the abstract to make it attractive to potential readers. We appreciate that not everyone would be interested in that part of the study and have taken your recommendation into account by summarizing the results further.

2. Abstract, line 7, lines 192f, 195, 202, 207, 222, 232, 348, 350, 353: "the probability (P) of observing a certain RHi" is mathematically unprecise as the following sentences are clearly describing the behaviour of the probability density. Specifically, "the probability of observing a certain RHi" is always zero as the boundaries for integrating the PDF are identical in this case. I suggest sticking to the terms "PDF" and "probability density" and not denoting the PDF with "P", as this is the common symbol for the probability.

We thank the Reviewer for this important remark. We have rewritten the corresponding sentences more precisely as recommended. We use "PDF" for discrete probability and "P" for continuous probabilities. For the sake of simplicity, we have deleted equation 5.

3. Abstract, line 19, lines 226, 308, 314, 325, 365, "decreasing pressure level": Unclear what you mean. Apparently, you sometimes mean increasing and sometimes decreasing pressure by that throughout the text. I suggest leaving out "level" and just using "increasing/decreasing pressure".

Modified as suggested. For the sake of clarity the word "level" has been removed. Only the expression "pressure increase/decrease" is now used.

4. Line 28: "saturated" "saturated with respect to ice"

Corrected

5. Lines 29, 130ff, 152, 179, 238, 271: I am sure what you mean is that the "water vapour" is ice supersaturated. "RHi" may rather be above 100% or above saturation.

We thank the Reviewer for the remark. We indeed mean water vapour is "ice supersaturated". We corrected this throughout the manuscript.

6. Lines 34f: In the common definition of supersaturation, 140% ice supersaturation would refer to RHi = 240%. Suggestion: "ice supersaturation" "RHi"

Modified as suggested.

7. Page 2, third paragraph: I missed a sentence on recent research suggesting a low efficacy for contrail cirrus potentially resulting in a low temperature response despite the high ERF (e.g. Bickel 2023).

We have added two sentences about this work at the end of the paragraph to flag that the efficacy of the RF by contrail and induced cirrus might be lower than 1. The new text can be found from line 45 to line 48 in the revised manuscript.

8. Line 51 f: An accurate representation of ISSRs is important for the prediction of contrail persistence, not "contrail-prone conditions" in general.

Modified as suggested.

9. Line 58: The way the sentence is written, it seems to state that ISSR are more frequent between 400 and 300 hPa in the midlatitudes and between 300 and 200 hPa in the high latitudes, which is not the case when looking at figure 9 in Lamquin et al. (2012). Please switch the order of either the pressure levels or the latitude regions.

Corrected.

10. Line 101: As the three pressure ranges appear numerous times in the paper, you may consider labelling them with something like "lower", "intermediate" and "upper" layer to improve the readability.

We thank the Reviewer for the recommendation. However, to avoid using too many labels, which could distract the readers, we have chosen not to label pressure layers.

11. Line 104: Is the number of measurements in the HL missing a digit or is it really an order of magnitude lower than in the other latitude regions on this pressure level ?

There is no missing digit in the number of measurements between 325 hPa and 275 hPa (Lower UTLS) in the HL. The density of flights in the ML is much higher and the region considered here as tropics is quite wide (see Fig.1), so that the number of measurements made in these two regions is greater than the number of measurements made in the high latitudes for the three pressure layers considered in the study.

12. Line 108: Why is this a good replacement? Are the quality flags of RHi and RHI correlated during periods, where the "grounding problem" did not exist?

The grounding problem occurred between the years 2011 and 2017. In fact, for now it is the quality flag of RHi which is not well derived. This flag is similar to that of RHI since RHi is deducted from RHI measurements. We have modified the related sentences as follows (from line 106 to line 109 in the revised manuscript): *It should be noted that, between the years 2011 and 2017, there was a grounding problem with the IAGOS data acquisition system. For this period, the quality flag of RHi is not well derived but it is known to be similar to that of RHI. We therefore selected RHi values using the RHI quality flag for this particular period.*

13. Lines 130f: Not RHi of the mixture but RHi in the ambient atmosphere has to be below 100% for non-persistent contrails. I suggest that you leave out "If RHi of the mixture is subsaturated," and simply state that contrails in general form under these conditions, as you mention the criterion for contrail persistence in the next sentence anyway. Also, please specify that you mean the properties of the ambient atmosphere by "T" and "RHI".

Modified as suggested.

14. Lines 156-160: Unclear description of fig. 3. I cannot reproduce the 30% for the 0.05 threshold in the tropics from the plots. Also, you mention two thresholds for the ML, but only give the percentages for one of them.

We thank the Reviewer for the remark. It is 40 % instead of 30 %.
We corrected it. We also gave the percentage for the two thresholds for the ML.

15. Line 191: Apparently, this is only true for the ML.

We agreed that the variations in RHi mean and standard deviation are substantial in the HL and in the tropical region. We have therefore modified the relevant sentence as follows: *In agreement with Reutter et al. (2020) who analyzed RHi in the North Atlantic, the mean and the standard deviation of RHi distributions in the UT vary little with the pressure level in the ML (Table 3). However, their variations are substantial in the tropical and HL regions (Table 3).* This piece of text stands from line 189 to line 191 in the revised manuscript.

16. Lines 236f, "It is the case here for the stratospheric PDF of RHi in the 325-275 hPa layer (b ≈0, Fig. 4a,e,…": Unclear. What is the case? Under

sampling? Also, b is clearly not zero for the stratosphere and figure 4e does not correspond to the mentioned layer.

We thank the Reviewer for this important remark. We indeed mean under-sampling. We also agree that the comment does not correspond to figure 4e and b is significantly different from zero for the stratosphere. We have therefore revised the concerned sentences as follows: *Our results show that this property is common to the RHi PDF in the ML and the HL of the Northern Hemisphere in the 275-225 and 225-175 hPa layers (Fig. 4b,c,e,f). For these pressure layers, Gierens et al. (1999) did not find the break in slope around 100 % in the MOZAIC data for the period 1995-1997. This might be due to an undersampling of the LS properties of RHi PDF over the period they considered. It is worth noting that in the 325-275 hPa layer, the break in slope around 100 % in the RHi PDF in the LS is marked only in the HL (Fig. 4a). In the ML, the break in slope seems to be undersampled in the USA in the MOZAIC and IAGOS data (Fig. 5a).* This piece of text stands from line 233 to line 238 in the revised manuscript.

17. Line 237f: Hard to verify this from the plots. The opposite seems to be the case for example for EU.

We thank the Reviewer for the remark. The regional disparities in RHi properties are also due to sampling differences between pressure layers (the 325-275 hPa layer is undersampled) and between regions (there are more measurements in the North Atlantic and the region considered as Europe is smaller). There are also differences due to variations in the latitude at which maximum sampling is performed. Consequently, statistics for this property are not robust. Since our aim is to document statistically robust properties of the PDF of RHi, we have removed the sentence concerned.

18. Figure 6: The PDF of in-cloud RHi increases for subsaturated conditions and decreases for supersaturated conditions when decreasing the detection threshold, indicating that the measurements that are added in this process were mainly sampled under subsaturated conditions, some of these under strongly subsaturated conditions, as even a "dry mode" appears in the PDFs. You briefly mention "detection uncertainty" in section 2.3, but never come back to this issue. Could these ice crystals in very dry air be crystals that have been turbulently mixed out of cirrus nearby or even false measurements? In both cases, these measurements would not really correspond to cloudy conditions. Please discuss your choice of threshold.

The choice of the threshold is now discussed on lines 273-280 in the revised manuscript as follows: *The PDF of in-cloud RHi decreases with RHi for subsaturated conditions but increases with RHi for supersaturated conditions when increasing the detection threshold (Fig. 6). This is consistent with the fact that, more generally, high concentrations of ice crystals are associated with high RHi (Petzold et al. 2017, Kramer et al. 2016). The lower tail of the PDF of RHi could include measurements in contrails or at the bottom of cirrus clouds where subsaturated conditions are observed more often (Dekoutsidis et al. 2023). They may also correspond to*

*measurements carried out in diluted cirrus clouds or in the proximity of cirrus clouds where ice crystals may be mixed with clear air by turbulence. Erroneous measurements may also contribute since the uncertainties associated with these thresholds are large (more than 50% for the threshold of 0.001 cm$^{-3}$ particles, see Section 2.3).*

Comments on the existence and origin of the dry mode in the in-cloud RHi PDF can be found between lines 265 and 266.

19. Line 306: I don't see, how the results for the other pressure layers are similar, as the frequencies are (almost) zero there, regardless of the fuel.

We have rephrased the comment in the revised manuscript (lines 324-325) as follows: *The impact of the fuel choice is much weaker for the 275-225 hPa and 225-175 hPa layers*.

20. Line 311f: Repetition of line 307f

Removed.

21. Line 318: Please be more precise what you mean. This is certainly only true when referring to the same pressure level. The tropical tropopause is usually colder than the tropopause in the ML and HL, but it is also located at a much lower pressure such that the tropical UTLS is not properly represented in your data set.

We think that the structure proposed in comment #22 below for the second sentence following the one referred to here, is sufficient to explain the dependence on the temperature of the impact of switching fuel. We therefore removed the sentence referred to here.

22. Page 19, first paragraph: Please rewrite the paragraph. Your argumentation is hard to follow. Try structuring it: ISSR + low ambient temperature  mixture of exhaust and ambient air exceeds water saturation due to curvature of vapour pressure curve, regardless of fuel.  persistent contrail ISSR + higher ambient temperature  mixture of exhaust and ambient air may stay subsaturated with respect to liquid water if ratio of EIH2O to Q is low.  no contrail with kerosene, persistent contrail with ethanol or hydrogen

The recommendation is taken into account. The new text can be found between lines 338 and 341 in the revised manuscript.

23. Line 330: "the properties of RHi"  "properties of the atmospheric RHi PDF"
Corrected.

24. Line 334: "RHi"  "RHi PDF"
Corrected.

25. Line 340f, "This frequency can be interpreted as the IAGOS aicraft flying time in cirrus clouds." This information belongs into the paragraph above, as this is not a result. Also. "estimated by" might be better than "interpreted as".

Modified as suggested.

26. Line 343: A dot is missing in "14 %". Anyway, I do not think this amount of details is necessary in a summary. "of the order of" implies that you give a rough estimate, but you repeat the exact numbers from the results section. I think, the interesting results are that cirrus is more frequent in the tropics than in the ML, that it occurs higher up in the tropics and that the frequency decreases if a higher detection threshold is chosen.

Modified as suggested.

27. Line 362: "In this region, aircraft form persistent contrails…" Suggestion: "The reason is that, in this region, aircraft running on kerosene already form persistent contrails…"

Modified as suggested.

28. Line 364: "contrail" "persistent contrail", also, the lower level in the tropics seems to be an exception from this rule.

We thank the reviewer for the remark. To report this exception, we modified the related sentence (line 382) as follows: "*It decreases with pressure between 325 and 175 hPa except in the tropics*"

**Technical Corrections:**

1. The formulation of the title seems a bit unwieldy to me. Suggestion: "Variability in the distribution properties of the relative humidity with respect to ice: Implications for contrail formation"

Modified as suggested.

2. Abstract, line 2, lines 89f: Several alternatives can be found on the internet for the meaning of the acronym MOZAIC. The IAGOS website itself gives the two possibilities "Measurement of OZone and water vapour on AIrbus in-service airCraft" and "Measurements of OZone and water vapour by in-service AIrbus airCraft". I suggest using one of these.

We thank the Reviewer for the useful remark. We used this one: "Measurement of OZone and water vapour on AIrbus in-service airCraft"

3. Abstract, line 15: I suggest starting a new paragraph after "large-scale simulations of RHi.".

We agree that we can start a new paragraph after "large-scale simulations of RHi". However, we prefer not to start a new paragraph, as a multi-paragraph abstract is not common.

4. Page 2, third paragraph: The last sentence seems a bit unconnected. I suggest inserting "also" after "Natural cirrus clouds".

We have modified the paragraph by placing the statement about natural cirrus clouds at the beginning of the paragraph. Please find it on lines 37-38 in the revised manuscript.

5. Line 87: Typo in "relative humidy"
Corrected.

6. Line 112: Typo in "comparaison"
Corrected.

7. Figure 2: The font size of "High-latitudes" appears to be larger than above the other two panels.
Corrected.

8. Figure 3: It would be easier to compare the two latitude regions if equal pressure ranges were plotted with the same line style.
Done as suggested.

9. Line 201: b is close to -4
Corrected.

10. Line 208: Typo in "Nord"
Corrected.

11. Table 3, caption: "RHi" is missing. Also, ML and HL have to be switched. Actually, this bracket is unnecessary anyway, as these abbreviations are not used in the table. The same holds for the caption of table 6.
Corrected.

12. Figure 4: "ML" is redundant in line 3 of the caption. Also, the font size of "Probability Density" appears to be larger in the last row of images. Furthermore, I would not call the plots of the last row "global", as they are clearly not.
The figure caption has been corrected. The size of the "Probability Density" is now uniform. Instead of calling the plot in the last row "global", we call it "HL + ML+ Tropis". We have also specified in the manuscript that we mean by "global RHi PDF" the RHi PDF combining measurements in HL, ML and Tropis.

13. Figure 5: Typo in the figure key: "LT"

Corrected.

14. Table 5: A dot is missing in the bottom right cell.

Corrected.

15. Line 252: The reference probably should be figure 4, not 5.

Corrected.

16. Line 253: The second "RHi" is redundant.

Corrected.

17. Line 257: Typo in "aformentioned"

Corrected.

18. Line 260: Typo in "completly"

Corrected.

19. Figure 6, caption: The "C" is missing behind the temperature value.

Corrected.

20. Figure 7, caption: Typo in "blye"

Corrected.

21. Figures 7 and 8: A grid or at least horizontal lines would improve the readability of the figures. Also, one might consider combining the two figures into one by plotting the non- persistent contrails on top of the persistent ones. This way, the frequency of non-contrail conditions could be read from the figure as well.

The grid lines are added as suggested. However, the impact of the fuel change on the frequency of persistent contrails is no longer sufficiently well visible if plotted with that of non-persistent contrails on the same figure. This is why we have plotted them separately.

22. Line 305: Typo in "hyrogen"

Corrected.

23. Line 342: "respective" is redundant.

Corrected.

24. Line 355: "saturated"  "supersaturated"

Corrected.

Reference:

Bickel, M. (2023). Climate impact of contrail cirrus (Doctoral dissertation, lmu).

---

## Referee Report (RR1)

**Comments on Sanogo et al. 2024, revised version**

- Abstract, line 3: Unfortunately, the name of the MOZAIC programme is still incorrect, as "airCraft" and "In-service" need to be switched.
- Line 131: Please remove "non-persistent". What you describe, is the condition for contrail formation in general, both, persistent AND non-persistent contrails.
- Line 133: Please move the sentence "Contrails are persistent if RHi is above 100% (See Fig. 3 in Schumann, 1996)." to the end of the paragraph, as this does not belong to the Schmidt-Appleman criterion. The SAc has nothing to do with persistence.
- Figure 3, caption: "number density" is missing after "ice crystal".
- Line 212f: This seems to be exactly the other way.
- Figure 5, key: "LT" -> "LS"
- Line 251f: The references to the two figures need to be switched.
- Line 273f: This makes no sense. Please remove "with RHi" two times in this sentence.
- Line 339: "The contrail plume…" -> "At higher ambient temperatures, the contrail plume…"
- Line 340: "In such conditions, …" -> "Therefore, …"

---

## Author Response (AR2)

Dear Editor,

Please find enclosed the new version of the manuscript and our response to the last round of Reviewer's comments. Before anything we would like to thank you and the Reviewers for the careful reviews, which clearly improved our work in form and in content.

Our answers are provided on a point-by-point basis and the following convention for text fonts is used:

- Reviews comment in black color
- Our answer in blue color

Comments on Sanogo et al. 2024, revised version

1. Abstract, line 3: Unfortunately, the name of the MOZAIC programme is still incorrect, as "airCraft" and "In-service" need to be switched.

Corrected

2. Line 131: Please remove "non-persistent". What you describe, is the condition for contrail formation in general, both, persistent AND non-persistent contrails.

Done

3. Line 133: Please move the sentence "Contrails are persistent if RHi is above 100% (See Fig. 3 in Schumann, 1996)." to the end of the paragraph, as this does not belong to the Schmidt-Appleman criterion. The SAc has nothing to do with persistence.

Done

4. Figure 3, caption: "number density" is missing after "ice crystal".

Done

5. Line 212f: This seems to be exactly the other way.

Done

6. Figure 5, key: "LT" -> "LS"

Corrected

7. Line 251f: The references to the two figures need to be switched.

Done

8. Line 273f: This makes no sense. Please remove "with RHi" two times in this sentence.

Done

9. Line 339: "The contrail plume…" -> "At higher ambient temperatures, the contrail plume…"

Done

10. Line 340: "In such conditions, …" -> "Therefore, …"

Done